

# Wide space sowing achieved high productivity and effective nitrogen use of irrigated wheat in South Shanxi, China

Qiang Wang[1,2], Hafeez Noor[1,2], Min Sun[1,2], Aixia Ren[1,2], Yu Feng[1,2], Peng Qiao[1,2], Jingjing Zhang[1,2] and Zhiqiang Gao[1,2]

[1] College of Agriculture, Shanxi Agricultural University, Taigu, Shanxi, China
[2] State Key Laboratory of Sustainable Dryland Agriculture, Shanxi Agricultural University, Taiyuan, Shanxi, China

Corresponding author
Zhiqiang Gao, gaosxau@163.com

## ABSTRACT

Wheat (*Triticum aestivum* L.) is a staple crop worldwide, and its yield has improved since the green revolution, which was attributed to chemical nitrogen (N) fertilizer application. However, regular N application decreases N use efficiency (NUE, the ratio of grain dry matter yield to N supply from soil and fertilizer). Various practices have been implemented to maintain high crop yield and improve NUE. Nowadays, the enhanced sowing method, *i.e.*, wide space sowing (WS), has improved the productivity of wheat crops. However, how the sowing method and N application rate affect N use and yield productivity has not been fully elucidated. Field experiments with treatments using two sowing methods (WS, and drill sowing, DS) and four N application rates (0, 180, 240, and 300 kg ha$^{-1}$, represented as N0, N180, N240, and N300, respectively) were conducted from 2017 to 2019. The results showed that grain yield under WS was 13.57–16.38% higher than that under DS. The yield advantage under WS was attributed to an increased ear number. Both the higher stem and productive stem percentage accounted for the increased ear number under WS. Higher total N quantity and larger leaf area index at anthesis under WS contributed to higher dry matter production, resulting in higher grain yield. Higher dry matter production was due to pre-anthesis dry weight and post-anthesis dry weight. The wheat crop under WS had a 12.44–15.00% higher NUE than that under DS. The increased NUE under WS was attributed to higher N uptake efficiency (the ratio of total N quantity at maturity to N supply from soil and fertilizer), which was the result of greater total N quantity. The higher total N quantity under WS was due to both higher pre-anthesis N uptake and post-anthesis N uptake. Remarkably, compared to DS with 240 kg N ha$^{-1}$, WS with 180 kg N ha$^{-1}$ had almost equal grain yield, dry matter, and total N quantity. Therefore, wheat crops under WS could achieve both high NUE and grain yield simultaneously with only moderate N fertilizer in South Shanxi, China.

## INTRODUCTION

With the current growth rate, the global population is expected to reach approximately 10 billion by 2050 (*United Nations Department of Economic & Social Affairs/Population*
*Division, 2017*). Increasing crop yields to maintain food security while reducing the environmental impacts of agriculture is a dual challenge for humans in the future (*Manschadi & Soltani, 2021*). Wheat is a staple food that feeds approximately 30% of the world's population (*Fahad, Abdul & Adnan, 2018*). China is the largest producer of wheat in the world. The North China Plain (NCP) is one of the most vital cereal production regions in China, accounting for 25% of national food production (*Duan et al., 2019*; *Fan et al., 2019*). Thus, future productivity of wheat will have a greater influence on China and global food security.

Nitrogen (N) is a major driver of crop production, as it directly influences the dry matter production of crop plants by influencing the leaf area, radiation interception, and photosynthetic efficiency (*Duan et al., 2019*; *Manschadi & Soltani, 2021*; *Li et al., 2022*). Crop yield and quality depend considerably on N application (*Zhang et al., 2016*). In many parts of the world, a large increase in N fertilizer input is required to increase crop yield (*Manschadi & Soltani, 2021*). However, the increase in crop yield did not match the increase in N fertilizer input. For example, from 1980 (9.3 Mt) to 2012 (24 Mt), a 158% increase in N fertilizer input was associated with a 70% increase in China's crop yield (321–547 Mt) (*Yang et al., 2017*). In addition, excessive N input significantly reduces crop yield and N use efficiency (NUE; the ratio of grain dry matter yield to N supply, which is the sum of N from soil and fertilizer) (*Nehe et al., 2020*; *Manschadi & Soltani, 2021*). Furthermore, excessive N input also leads to N fertilizer residue, which causes many environmental problems, such as soil acidification, $N_2O$ emissions, and decreased soil microbial activity (*Zhang et al., 2015*; *Duan et al., 2019*). Therefore, it is widely recognized that improving NUE can alleviate hazards to the environment of crop production systems and improve their economic and environmental performance (*Yang et al., 2017*).

NUE can be defined as the grain dry matter yield (kg ha$^{-1}$) divided by the supply of available N from soil and fertilizer (kg N ha$^{-1}$; *Moll, Kamprath & Jackson, 1982*). NUE was calculated as the product of two subcomponents: (i) N uptake efficiency (total N at maturity/N supply from soil and fertilizer; NUpE), and (ii) N utilization efficiency (grain dry matter yield/total N quantity at maturity; NUtE). In addition to NUE and its components, agronomic N use efficiency ($AE_N$), N recovery efficiency ($RE_N$), and partial factor productivity of applied N ($PFP_N$) have been used to evaluate the efficiency of N use. $AE_N$ is defined as the difference in grain yield in the N treatment minus the grain yield in the blank N treatment divided by the N supply from the N fertilizer, which indicates grain yield produced per unit of supplied N fertilizer (*Zhang et al., 2015*). $RE_N$ is defined as the difference in total N in the N treatment minus the total N in the blank N treatment divided by the N supply from the N fertilizer, which indicates the percentage of fertilizer N absorbed by plants (*Yang et al., 2017*). $PFP_N$ is the ratio of grain yield to the fertilizer N supplied, which indicates the grain yield produced per unit of fertilizer applied (*Cox, Qualset & Rains, 1986*). According to the current agricultural production situation, N fertilizer input is a common management strategy to achieve high crop yields (*Duan et al., 2019*; *Li et al., 2022*). Increasing N fertilizer application can significantly improve crop yields but inevitably reduce NUE according to the above definition of N use-related efficiencies (*Chen et al., 2016*; *Yang et al., 2017*). It may seem impossible to achieve both a

high yield and NUE simultaneously. Nonetheless, solving this issue is crucial to enhance grain production.

An important factor in improving wheat yield is the sowing method, which influences the spatial distribution and growth of plants (*Fan et al., 2019*; *Liu et al., 2020*). Compared to the traditional sowing method, drill sowing (DS), the wide space sowing (WS) alters the former sowing width from 2–3 cm to 5–8 cm in addition to changing the seed distribution by separating single grains from each other instead of planting all the seeds in a line while using the same seeding rate (*Zhao et al., 2013*). It was reported that a high winter grain yield of 12.4 t ha$^{-1}$ was achieved under WS in North China (*Liu et al., 2020*). Other studies have added further evidence to indicate that WS is useful for improving crop productivity in China (*Fan et al., 2019*; *He, 2020*). *Liu et al. (2017)* showed that the ear number of wheat under WS was significantly higher than that under DS, accounting for the greater grain yield under WS than DS. It has been reported that the increased ear number is attributed to the stem number rather than the productive stem percentage (*Chu et al., 2018*). It has also been reported that wheat plant under WS had a higher NUE than that under DS (*Chu et al., 2018*; *Liu et al., 2021a*). However, it remains unclear whether WS can achieve both high yield and NUE simultaneously.

South Shanxi is located in the NCP and supplies more than 50% of the winter wheat produced in China. However, excessive N application is common in this region, which causes decreased NUE. Considering the demand for achieving a high yield, high NUE and environmental protection are urgently needed in this region. Thus, we conducted a 2-year field experiment to examine the effects of sowing method and N application on winter wheat production and NUE. The main objectives of this study were to (1) clarify the effects of N application rate under WS on population development and yield formation, and (2) determine whether WS could help improve both yield and NUE simultaneously.

## MATERIALS AND METHODS

### Site description

Field experiments were conducted in a farmer's field in Shangyuan Village, Hougong Township, Wenxi County, Shanxi Province, China (35°24′N, 111°26′E) in 2017–2018 and repeated in a nearby field in 2018–2019. This site has a typical semi-arid warm temperature and continental monsoon climate (Köppen classification) with an average daily temperature of 8.6 °C, average precipitation of 190.5 mm, and 3015.6 MJ m$^{-2}$ of total solar radiation during the wheat growing season (from mid–October to early June) from 2005 to 2015. Soil samples from the upper 20 cm layer were randomly collected with five replicates for soil analysis before wheat was sown in 2017 and 2018. The soil type was classified as calcareous cinnamon soil according to Chinese soil taxonomy with a pH of 8.47–8.61, organic matter of 13.61–14.31 g kg$^{-1}$, total N of 1.01–1.05 g kg$^{-1}$, alkaline N of 40.05–44.07 mg kg$^{-1}$, Olsen P of 10.71–11.25 mg kg$^{-1}$, and available K of 188.87–200.24 mg kg$^{-1}$ in 2017–2018 and 2018–2019 (Table 1). The cropping pattern of the experimental site is a winter wheat–summer maize double–cropping system. The climate parameters in 2017–2018 and 2018–2019 were collected from a weather station (Watchdog 2000 Series;

**Table 1 Soil basic fertility before sowing at the experimental site in 2017–2018 and 2018–2019.**

| Year | pH | Organic matter (g kg$^{-1}$) | Total N (g kg$^{-1}$) | Alkaline N (mg kg$^{-1}$) | Olsen P (mg kg$^{-1}$) | Available K (mg kg$^{-1}$) |
|---|---|---|---|---|---|---|
| 2017–2018 | 8.61 | 13.61 | 1.06 | 44.07 | 10.71 | 188.87 |
| 2018–2019 | 8.47 | 14.31 | 1.01 | 40.05 | 11.25 | 200.24 |

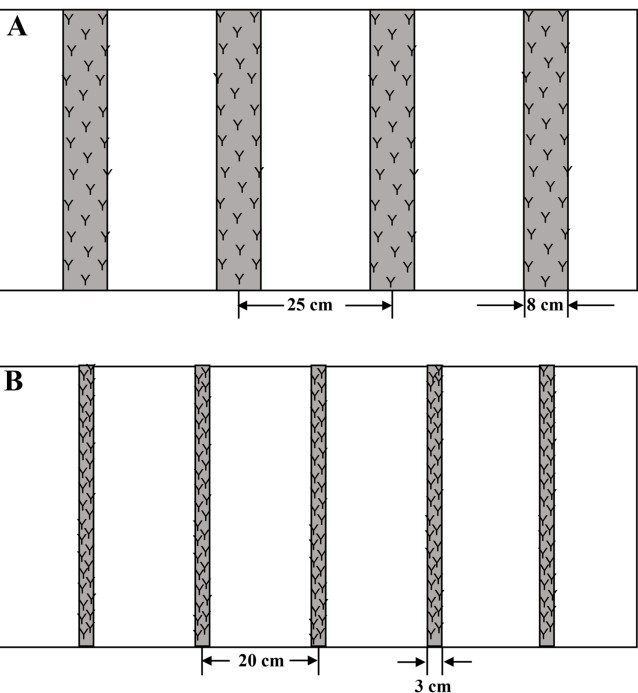

**Figure 1 The sketch maps of wide space sowing (A) and drill sowing (B) used in this study.** The plant densities at the three-leaf stage were 295 and 312 plant m-2 in 2017 and 2018, which was quite same under the two sowing methods.

Spectrum Technologies Inc, Aurora, IL, USA) approximately 200 m from the experimental field.

## Experimental design and crop management

The experiment was arranged in a split-plot design with the sowing method as the main plot and N application rate as a subplot with three replicates. Two sowing methods (Fig. 1), wide space sowing (WS, sowing and row width were 8 and 25 cm, respectively) and drill sowing (DS, sowing and row width were 3 and 20 cm, respectively), and four N application rates, 0, 180, 240, and 300 kg ha$^{-1}$ (represented as N0, N180, N240, and N300, respectively) were applied in this study. DS was performed using a drill sowing machine (2BXF-12; Nonghaha Mechanical Co. Ltd., Shijiazhuang, China), and WS was performed using a wide space sowing machine (2BMYF-10/5; Yuncheng Gongli Co. Ltd., Heze, China).

Each subplot was 8 m in length and 4 m in width. A widely planted wheat cultivar, Liangxing99, was planted on 25[th] October and 11[th] October 2017 and 2018. The expected

plant density was approximately 300 plants m$^{-2}$ for both sowing methods. The plant densities at the three-leaf stage (Zadoks code 13) were 295 and 312 plants m$^{-2}$ in 2017 and 2018, respectively, and there was no significant difference between WS and DS.

N fertilizer was applied as urea (46% N), with 60% of the N fertilizer being applied before sowing, and 40% as topdressing fertilizer during jointing (Zadoks code 32); 150 kg P$_2$O$_5$ ha$^{-1}$ in the form of calcium super-phosphate (16% P$_2$O$_5$) and 90 kg K$_2$O ha$^{-1}$ in the form of potassium chloride (52% K$_2$O) were applied before sowing. Each plot was irrigated thrice, with 60 mm (1.92 m$^3$/plot) water at wintering (Zadoks code 26), jointing (Zadoks code 32), and anthesis (Zadoks code 65). Irrigation water was supplied by a movable sprinkler system, and the amount of water applied was measured using a flow meter. The field was kept free of diseases, pests, insects, and using pesticides as needed. Weeds were well-controlled with herbicide administration two or three times in each experimental year.

## Sampling and measurements

The stem number (sum of main stems and tillers) of wheat plants was counted in a typical and central row of 1 m length at jointing, at which the wheat plants had the maximum stems number (*Lu et al., 2021*). The productive stem percentage was calculated as the ratio of ear number to the maximum stem number. During each growing season, the wheat plants were sampled in a row of 0.5 m length at anthesis and maturity (Zadoks code 91). At anthesis, all green leaves were separated and measured using a leaf area meter (LI–3100C, LI–COR, Lincoln, NE, USA) to calculate the leaf area index. All samples were then divided into ear and vegetative parts (stems, sheaths plus leaves). At maturity, after counting the ear number, the samples were divided into grain and straw (stems, sheaths, leaves, chaff plus rachis). All separated samples were oven dried at 105 °C for 30 min and weighed after further drying at 70 °C to a constant weight. The grain number per ear and 1,000-grain weight were calculated using the grain samples described above. The yield was determined from a 10 m$^2$ area at maturity in the center of each plot and adjusted to a standard moisture content of 0.125 g H$_2$O g$^{-1}$ fresh weight. The grain moisture content was measured using a digital moisture tester (PM8188A; Kett Electric Laboratory, Tokyo, Japan).

After the dry matters of all separated sample plants at anthesis and maturity were weighed, all the samples were shredded using a plant ball mill pulverize (Jxfstprp-11; Jingxin Co Ltd, Shanghai, China) for N concentration measurement. The N concentration in the samples was determined using the standard indophenol-blue colorimetric method (*Novamsky et al., 1974*).

Soil samples were collected from 0–20, 20–40, 40–60, 60–80, and 80–100 cm soil depths in each plot, and were analyzed for total mineral N content (NO$_3^-$-N and NH$_4^+$-N) using the method described by *Wagner (1974)* and *Benesch & Mangelsdorf (1972)*.

## Statistical analysis and calculations

The experimental data were statistically analyzed using Microsoft Excel 2016 and Statistix 8.0 (Analytical Software, Tallahassee, FL, USA), and figures were generated using Origin Lab Pro 2021b (OriginLab Corporation, Northampton, MA, USA). All data are presented

as the means of three replicates ($n = 3$). Comparisons among multiple groups were performed using Tukey's honestly significant difference (HSD) test. Differences were considered statistically significant at $P < 0.05$. Statistix 8.0 software was used for the variance analysis.

The accumulation, partitioning, and translocation of dry matter and N were calculated using the following equations (*Laza et al., 2003*; *Cox, Qualset & Rains, 1986*):

$$\text{Post−anthesis dry matter production } (\text{DM}_{\text{post}}, \text{t ha}^{-1}) \\ = \text{total dry weight at maturity} − \text{TDW}_{\text{as}} \tag{1}$$

$$\text{Harvest index } (\%) = \frac{\text{grain dry weight}}{\text{total dry weight at maturity}} \tag{2}$$

$$\text{Post−anthesis accumulated N } (\text{N}_{\text{post}}, \text{kg ha}^{-1}) = \text{TN} − \text{TN}_{\text{as}} \tag{3}$$

$$\text{N harvest index } (\text{NHI}, \%) = \text{GN/TN} \tag{4}$$

where $\text{TDW}_{\text{as}}$ (t ha$^{-1}$) is the total dry weight at anthesis. TN (kg ha$^{-1}$) and $\text{TN}_{\text{as}}$ (kg ha$^{-1}$) are the total N quantity at maturity and anthesis, respectively. GN (kg ha$^{-1}$) is grain N content.

N use-related traits were calculated using the following equations (*Moll, Kamprath & Jackson, 1982*; *Foulkes et al., 2009*):

$$\text{N uptake efficiency } (\text{NUpE}, \%) = \text{TN/soil N (pre−sowing soil mineral N} + \text{N}_{\text{f}}) \tag{5}$$

$$\text{N utilisation efficiency } (\text{NUtE}, \text{kg kg}^{-1}) = \text{grain dry weight/TN} \times 1{,}000 \tag{6}$$

$$\text{N use efficiency } (\text{NUE}, \text{kg kg}^{-1}) = \text{NUpE} \times \text{NUtE} \tag{7}$$

$$\text{Agronomic N use efficiency } (\text{AE}_{\text{N}}, \text{kg kg}^{-1}) = (\text{Y}_{\text{N}} − \text{Y}_0)/\text{N}_{\text{f}} \times 1{,}000 \tag{8}$$

$$\text{N recovery efficiency } (\text{RE}_{\text{N}}, \%) = (\text{TN}_{\text{N}} − \text{TN}_0)/\text{N}_{\text{f}} \tag{9}$$

$$\text{Partial factor productivity of applied N } (\text{PFPN}, \text{kg kg}^{-1}) = \text{Yield/N}_{\text{f}} \times 1{,}000 \tag{10}$$

where $\text{Y}_{\text{N}}$ and $\text{Y}_0$ are the yield (t ha$^{-1}$) in the N fertilization and N0 treatments, respectively. $\text{TN}_{\text{N}}$ and $\text{TN}_0$ are the total N quantity at maturity (kg ha$^{-1}$) in the N fertilization and N0 treatments, respectively. $\text{N}_{\text{f}}$ is the total input of N fertilizer (kg ha$^{-1}$).

# RESULTS

## Weather conditions and crop growth duration

Seasonal precipitation was 51.2 mm greater in 2017–2018 than that in 2018–2019 due to the greater rainfall from jointing to booting and from anthesis to maturity in the former growing season (Table 2). However, there was a higher mean temperature, more accumulated temperature, and greater incident solar radiation during all growing periods

**Table 2 The climate parameters in 2017–2018 and 2018–2019.**

| Year | Period | Precipitation mm | Daily mean temperature °C | Total accumulated temperature °C d | Incident radiation MJ m$^{-2}$ |
|---|---|---|---|---|---|
| 2017–2018 | S-J | 35.3 | 3.2 | 654.7 | 1395.1 |
| | J-B | 47.5 | 11.9 | 226.1 | 303.9 |
| | B-A | 27.1 | 18.1 | 344.7 | 360.8 |
| | A-M | 44.3 | 20.9 | 753.1 | 708.1 |
| | Whole season | 154.2 | 7.9 | 1,978.6 | 2,767.9 |
| 2018–2019 | S-J | 48.9 | 3.7 | 732.8 | 1,588.0 |
| | J-B | 20.7 | 13.2 | 304.5 | 338.1 |
| | B-A | 22.7 | 18.6 | 371.4 | 370.0 |
| | A-M | 10.7 | 21.8 | 871.6 | 738.5 |
| | Whole season | 103.0 | 8.7 | 2,280.3 | 3,034.5 |

Note:
S-J, from sowing to jointing; J-B, from jointing to booting; B-A, from booting to anthesis; A-M, from anthesis to maturity.

**Table 3 Growth durations in winter wheat growing season in 2017–2018 and 2018–2019.**

| Year | S-J | J-B | B-A | A-M | Whole season (d) |
|---|---|---|---|---|---|
| 2017–2018 | 155 | 19 | 19 | 36 | 229 |
| 2018–2019 | 162 | 23 | 20 | 40 | 245 |

Note:
S-J, from sowing to jointing; J-B, from jointing to booting; B-A, from booting to anthesis; A-M, from anthesis to maturity.

in 2018–2019 than those in 2017–2018, especially from jointing to booting and from anthesis to maturity.

The growing durations in each period (sowing to jointing, jointing to booting, and anthesis to maturity) were longer in 2018–2019 than those in 2017–2018; thus, the total growing durations were longer in 2018–2019 (Table 3). It should be noted that the wheat crop was sowed late (by approximately 10 days) in 2017–2018 because of the continuous rainy weather before sowing.

## Yield and yield related attributes

The sowing method and N application rate significantly affected grain yield (Table 4). The wheat crop under WS produced higher grain yield by 16.38% and 13.57%, averaged across N application rates, than that under DS in 2017–2018 and 2018–2019, respectively. There were significant increases in yield when the N application rate increased from 0 to 180 kg ha$^{-1}$ and then to 240 kg ha$^{-1}$ under both sowing methods and during the two growing seasons. The sowing method and N application rate had a significant combined effect on grain yield. Grain yield improved slightly with an increase in the N application rate (from 240 to 300 kg ha$^{-1}$) under WS in both growing seasons, whereas significant yield reductions (11.78% in 2017–2018 and 7.21% in 2018–2019) were observed under DS (lodging occurred during grain filling under DS with 300 kg N ha$^{-1}$ applied). In addition, the wheat crop under WS treated with 180 kg ha$^{-1}$ N produced a commensurate yield compared with that under DS with 240 kg N ha$^{-1}$ applied in both growing seasons.

**Table 4 Grain yield and yield components of winter wheat under different sowing method and N treatment in 2017–2018 and 2018–2019.**

| Year | Sowing method | N treatment | Yield (t ha$^{-1}$) | Ear number ($10^6$ ha$^{-1}$) | Grain number per ear | 1,000-grain weight (g) |
|---|---|---|---|---|---|---|
| 2017–2018 | DS | N0 | 3.88 d | 2.59 d | 28.5 c | 44.1 a |
| | | N180 | 6.31 b | 4.08 c | 33.3 a | 41.2 b |
| | | N240 | 6.79 a | 4.33 b | 33.2 a | 40.9 b |
| | | N300 | 5.99 c[#] | 4.53 a | 30.4 b | 38.2 c |
| | | Mean | 5.74 B | 3.88 B | 31.3 A | 41.1 A |
| | WS | N0 | 4.43 c | 3.41 d | 25.8 c | 43.0 a |
| | | N180 | 6.86 b | 4.60 c | 32.9 a | 40.1 b |
| | | N240 | 7.64 a | 5.01 b | 32.7 a | 40.0 b |
| | | N300 | 7.79 a | 5.65 a | 29.9 b | 39.1 c |
| | | Mean | 6.68 A | 4.67 A | 30.3 B | 40.6 A |
| 2018–2019 | DS | N0 | 5.02 c | 3.56 d | 27.0 c | 44.6 a |
| | | N180 | 7.47 b | 4.77 c | 32.8 a | 42.0 b |
| | | N240 | 8.04 a | 5.19 b | 32.8 a | 41.9 b |
| | | N300 | 7.46 b[#] | 5.49 a | 29.4 b | 40.2 c |
| | | Mean | 7.00 B | 4.75 B | 30.5 A | 42.2 A |
| | WS | N0 | 5.51 c | 4.30 d | 24.8 c | 44.3 a |
| | | N180 | 8.23 b | 5.36 c | 31.7 a | 42.3 b |
| | | N240 | 8.95 a | 6.10 b | 30.8 a | 41.9 b |
| | | N300 | 9.12 a | 6.82 a | 28.9 b | 40.4 c |
| | | Mean | 7.95 A | 5.65 A | 29.0 B | 42.2 A |
| ANOVA | | | | | | |
| Year (Y) | | | ** | ** | ** | ** | ** |
| Sowing (S) | | | ** | ** | ** | ** | ns |
| N rate (N) | | | ** | ** | ** | ** | ** |
| Y × S | | | ns | ns | ns | ns | ns |
| Y × N | | | ns | ns | ns | ns | ns |
| S × N | | | ** | ** | ** | ** | ** |
| Y × S × N | | | ** | ** | ** | ** | ** |

**Notes:**
[#] Lodging occurred during grain filling.
** Significant at 0.01 probability level.
DS, drill sowing; WS, wide space sowing. Within a column for each growing season, means followed by different uppercase letters are significantly different according to Tukey's HSD test ($\alpha = 0.05$) between two sowing methods. Within a column for sowing method, means followed by different lowercase letters are significantly different according to Tukey's HSD test ($\alpha = 0.05$) among four N treatments. ns, not significant at 0.05 probability level.

The positive effect of WS on grain yield was mainly because of the increased ear number per hectare (Table 4). Averaged across N application rates, the wheat crop under WS exhibited a 20.36% and 18.95% higher ear number per hectare than that under DS in 2017–2018 and 2018–2019, respectively. There was little or no difference in the grain number per ear and 1,000-grain weight between sowing methods. With an increase in the N application rate, although it was observed that the ear number increased under both sowing methods. The rates of ear number increased under WS with improved N application rate (from 180 to 240 kg ha$^{-1}$ and from 240 to 300 kg ha$^{-1}$) were significantly

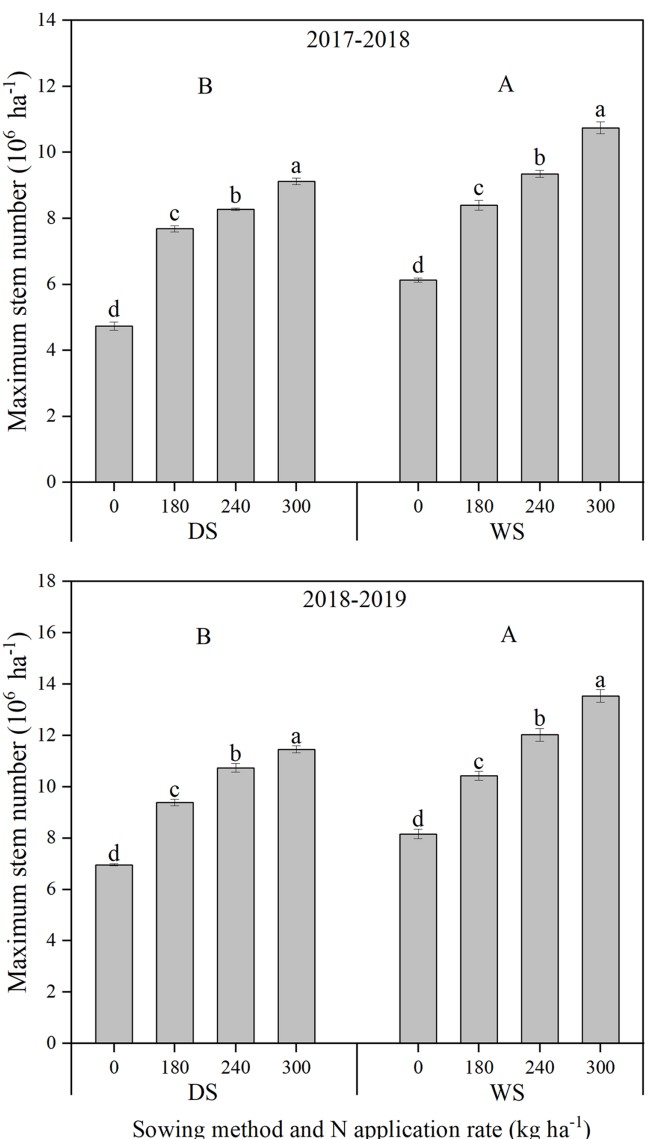

**Figure 2 Maximum stem number of winter wheat under different sowing method and N application rate in 2017–2018 and 2018–2019.** DS, drill sowing; WS, wide space sowing. Within each growing season, different uppercase letters are significantly different according to Tukey's HSD test ($\alpha = 0.05$) between two sowing methods. Within each growing season and sowing method, different lowercase letters above bars are significantly different according to Tukey's HSD test ($\alpha = 0.05$) among four N application rates. error bar, the standard deviation (± SD).

larger than those under DS (8.91–13.81% *vs*. 4.62–8.81%). However, with an increase in the N application rate, a decreasing trend in grain number per ear and 1,000-grain weight was recorded under both sowing methods and in the two growing seasons.

The maximum stem number and productive stem percentage under WS were both significantly higher than those under DS (Figs. 2 and 3). The improvement in maximum stem number (14.52–16.09%), rather than productive stem percentage (3.24–3.85%), mainly accounted for the wheat crop under WS producing more ears per hectare. With an increase in the N application rate, the maximum stem number of winter wheat

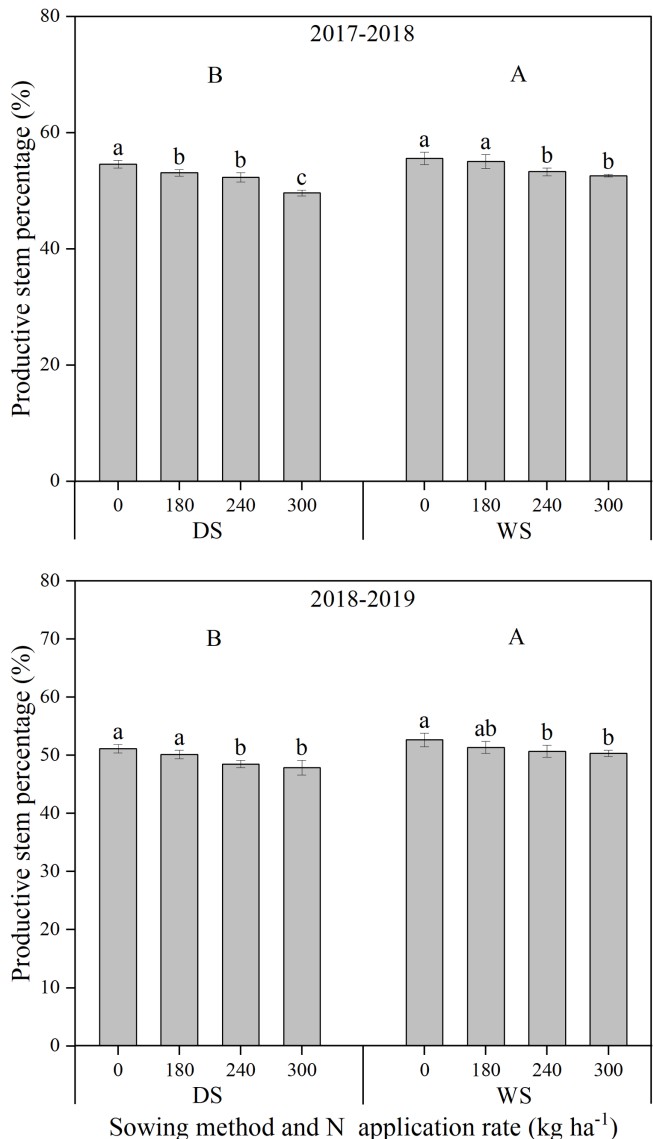

**Figure 3 Productive stem percentage of winter wheat under different sowing method and N application rate in 2017–2018 and 2018–2019.** DS, drill sowing; WS, wide space sowing. Within each growing season, different uppercase letters are significantly different according to Tukey's HSD test ($\alpha = 0.05$) between two sowing methods. Within each growing season and sowing method, different lowercase letters above bars are significantly different according to Tukey's HSD test ($\alpha = 0.05$) among four N application rates. error bar, the standard deviation (± SD).

significantly and continuously increased. However, a decreasing trend was recorded in the productive stem percentage under both sowing methods and in the two growing seasons.

The leaf area index at anthesis under WS was significantly higher by 6.70% and 7.97%, averaged across N application rates, than that under DS in 2017–2018 and 2018–2019, respectively (Fig. 4). With an increase in the N application rate, the leaf area index at anthesis significantly increased under both sowing methods and in the two growing seasons.

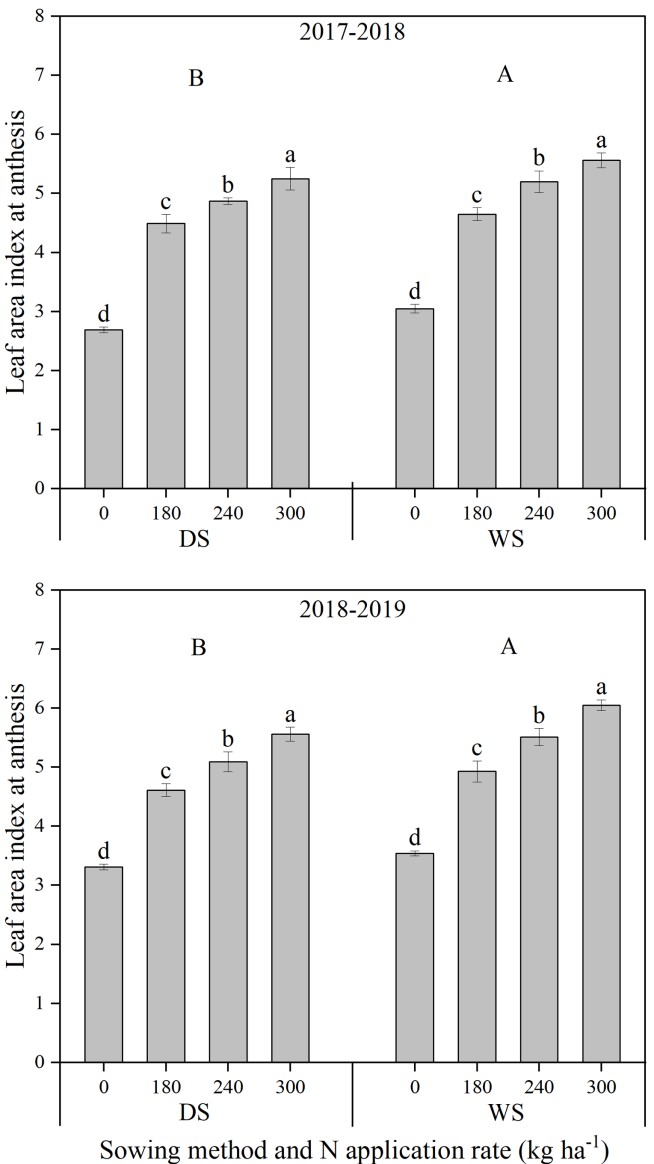

**Figure 4 Leaf area index at anthesis of winter wheat under different sowing method and N application rate in 2017–2018 and 2018–2019.** DS, drill sowing; WS, wide space sowing. Within each growing season, different uppercase letters are significantly different according to Tukey's HSD test ($\alpha = 0.05$) between two sowing methods. Within each growing season and sowing method, different lowercase letters above bars are significantly different according to Tukey's HSD test ($\alpha = 0.05$) among four N application rates. error bar, the standard deviation ($\pm$ SD).

The total dry weight at maturity under WS was significantly higher by 13.62% and 15.26%, averaged across N application rates, than that under DS in 2017–2018 and 2018–2019, respectively, whereas no significant difference was observed in the harvest index between sowing methods (Table 5). The total dry weight at maturity significantly increased when the N application rate increased from 0 to 180 kg ha$^{-1}$ and then to 240 kg ha$^{-1}$ under both sowing methods and in the two growing seasons. With a further increase

**Table 5 Total dry weight, harvest index, total dry weight at anthesis, post-anthesis dry matter production of winter wheat under different sowing method and N treatment in 2017–2018 and 2018–2019.**

| Year | Sowing method | N treatment | Total dry weight (t ha$^{-1}$) | Harvest index (%) | Total dry weight at anthesis (t ha$^{-1}$) | Post-anthesis dry matter production (t ha$^{-1}$) |
|---|---|---|---|---|---|---|
| 2017–2018 | DS | N0 | 6.34 c | 51.2 a | 4.12 c | 2.22 c |
| | | N180 | 12.09 b | 46.3 b | 8.05 b | 4.04 b |
| | | N240 | 12.94 a | 45.8 b | 8.60 a | 4.34 a |
| | | N300 | 12.10 b | 43.4 c | 8.14 b | 3.96 b |
| | | Mean | 10.87 B | 46.7 A | 7.23 B | 3.64 B |
| | WS | N0 | 7.61 c | 49.7 a | 4.90 d | 2.71 c |
| | | N180 | 12.98 b | 46.6 b | 8.26 c | 4.72 b |
| | | N240 | 14.20 a | 46.0 b | 9.01 b | 5.19 a |
| | | N300 | 14.63 a | 45.1 b | 9.40 a | 5.23 a |
| | | Mean | 12.35 A | 46.9 A | 7.89 A | 4.46 A |
| 2018–2019 | DS | N0 | 9.27 c | 46.3 a | 6.09 c | 3.18 c |
| | | N180 | 14.90 b | 43.7 b | 9.77 b | 5.13 b |
| | | N240 | 16.46 a | 43.5 b | 10.72 a | 5.74 a |
| | | N300 | 15.20 b | 42.7 b | 10.04 b | 5.16 b |
| | | Mean | 13.96 B | 44.1 A | 9.16 B | 4.80 B |
| | WS | N0 | 10.19 c | 46.3 a | 6.53 c | 3.66 c |
| | | N180 | 16.90 b | 42.5 b | 10.91 b | 5.99 b |
| | | N240 | 18.53 a | 42.5 b | 11.83 a | 6.70 a |
| | | N300 | 18.75 a | 42.4 b | 12.06 a | 6.69 a |
| | | Mean | 16.09 A | 43.4 A | 10.33 A | 5.76 A |
| ANOVA | | | | | | |
| Year (Y) | | | ** | ** | ** | ** |
| Sowing (S) | | | ** | ns | ** | ** |
| N rate (N) | | | ** | ** | ** | ** |
| Y × S | | | ns | ns | ns | ns |
| Y × N | | | ns | ns | ns | ns |
| S × N | | | ** | ** | ** | ** |
| Y × S × N | | | ** | ** | ** | ** |

Notes:
** Significant at 0.01 probability level.
DS, drill sowing; WS, wide space sowing. Within a column for each growing season, means followed by different uppercase letters are significantly different according to Tukey's HSD test (α = 0.05) between two sowing methods. Within a column for sowing method, means followed by different lowercase letters are significantly different according to Tukey's HSD test (α = 0.05) among four N treatments. ns, not significant at 0.05 probability level.

in the N application rate (from 240 to 300 kg ha$^{-1}$), the total dry weight at maturity under WS slightly improved in both growing seasons, but significant reductions of 6.49% and 7.65% were observed under DS (lodging occurred during grain filling under DS with 300 kg N ha$^{-1}$ applied) in 2017–2018 and 2018–2019, respectively. The harvest indices with N fertiliser application were significantly lower than those without N application. There was no significant difference in the harvest index among treatments with N application, except that the N application rate of 300 kg ha$^{-1}$ resulted in significantly lower

values than N application rates of 180 and 240 kg ha$^{-1}$ under DS (lodging occurred during grain filling under DS with 300 kg N ha$^{-1}$ applied) in 2017–2018.

Total dry weight at anthesis and post-anthesis dry matter production were both significantly higher under WS than those under DS in the two growing seasons, which accounts for the advantage in total dry weight of the wheat crop under WS over DS (Table 5). Pre-anthesis dry matter production and post-anthesis dry matter production significantly increased when the N application rate increased from 0 to 180 kg ha$^{-1}$ and then to 240 kg ha$^{-1}$ under both sowing methods and in the two growing seasons. When the N application rate further increased (from 240 to 300 kg ha$^{-1}$), pre-anthesis dry matter production and post-anthesis dry matter production under WS exhibited an increasing or unchanging trend in both growing seasons, but significant reductions of 5.35–6.34% and 8.76–10.10% were recorded under DS (lodging occurred during grain filling under DS with 300 kg N ha$^{-1}$ applied) in 2017–2018 and 2018–2019, respectively.

## N uptake, utilization, and related efficiencies

There was no significant difference in grain N concentration between the sowing methods, averaged across N application rates, in the two growing seasons (Table 6). Among the N treatments, there was no significant difference in grain N concentration under DS. However, significant improvements (5.02–5.98%) were recorded at 300 kg N ha$^{-1}$ under WS in the two growing seasons. The grain N content under WS was significantly higher than that under DS by 18.67% and 15.24% in 2017–2018 and 2018–2019, respectively. The grain N content significantly increased when the N application rate increased from 0 to 180 kg ha$^{-1}$ and then to 240 kg ha$^{-1}$ under both sowing methods and in the two growing seasons. When the wheat crop received more N fertiliser (300 kg ha$^{-1}$), the grain N content under WS significantly and continuously improved in both growing seasons; however, significant reductions of 9.78% and 6.22% were observed under DS (lodging occurred during grain filling under DS with 300 kg N ha$^{-1}$ applied) in 2017–2018 and 2018–2019, respectively.

The wheat crop under WS had a higher total N quantity at maturity and N harvest index, averaged across N application rates, than those under DS in both growing seasons (Table 6). With an increase in the N application rate, the total N quantity at maturity significantly and consistently increased under both sowing methods and in the two growing seasons, whereas the N harvest index exhibited a decreasing trend. Pre-anthesis N uptake and post-anthesis N uptake were both significantly higher under WS than those under DS in the two growing seasons, which accounts for the advantage in total N quantity at maturity under WS over DS. With an increase in the N application rate, both pre-anthesis N uptake and post-anthesis N uptake significantly and continuously increased under both sowing methods and in the two growing seasons. In addition, total N (including pre-anthesis N uptake and post-anthesis N uptake, and total N quantity at maturity) under DS with 240 kg N ha$^{-1}$ applied was commensurate with that under WS with 180 kg N ha$^{-1}$ applied.

The wheat crop under WS exhibited a higher NUE by 15.00% and 12.44%, averaged across N rates, than that under DS in 2017–2018 and 2018–2019, respectively (Table 7).

**Table 6 Grain N concentration and content (GNC% and GN) at maturity, total N quantity at maturity (TN), N harvest index (NHI), total N quantity at anthesis (TN$_{as}$), post-anthesis accumulated N (N$_{post}$) of winter wheat under different sowing method and N treatment in 2017–2018 and 2018–2019.**

| Year | Sowing method | N treatment | GNC% (%) | GN (kg ha$^{-1}$) | TN (kg ha$^{-1}$) | NHI (%) | TN$_{as}$ (kg ha$^{-1}$) | N$_{post}$ (kg ha$^{-1}$) |
|---|---|---|---|---|---|---|---|---|
| 2017–2018 | DS | N0 | 2.19 b | 71.3 c | 99.1 d | 71.9 a | 70.2 d | 28.9 d |
| | | N180 | 2.46 a | 137.6 b | 198.8 c | 69.2 b | 149.6 c | 49.2 c |
| | | N240 | 2.52 a | 149.3 a | 218.6 b | 68.3 b | 161.1 b | 57.5 b |
| | | N300 | 2.57 a | 134.7 b | 235.6 a | 57.2 c | 168.6 a | 67.0 a |
| | | Mean | 2.43 A | 123.2 B | 188.0 B | 66.6 B | 137.4 B | 50.6 B |
| | WS | N0 | 2.19 c | 82.6 d | 111.1 d | 74.4 a | 79.2 d | 31.9 d |
| | | N180 | 2.54 b | 153.6 c | 218.1 c | 70.4 b | 157.0 c | 61.2 c |
| | | N240 | 2.59 b | 169.0 b | 244.5 b | 69.1 b | 172.3 b | 72.2 b |
| | | N300 | 2.72 a | 179.5 a | 269.8 a | 66.5 c | 189.6 a | 80.2 a |
| | | Mean | 2.51 A | 146.2 A | 210.9 A | 70.1 A | 149.5 A | 61.4 A |
| 2018–2019 | DS | N0 | 2.12 b | 90.9 c | 123.6 d | 73.6 a | 91.6 d | 32.0 d |
| | | N180 | 2.30 a | 150.0 b | 222.9 c | 67.3 b | 156.3 c | 66.6 c |
| | | N240 | 2.31 a | 165.6 a | 247.5 b | 66.9 b | 172.6 b | 74.9 b |
| | | N300 | 2.39 a | 155.3 b | 264.2 a | 58.8 c | 179.6 a | 84.6 a |
| | | Mean | 2.28 A | 140.4 B | 214.6 B | 66.6 B | 150.0 B | 64.5 B |
| | WS | N0 | 2.12 c | 100.0 d | 136.0 d | 73.5 a | 99.0 d | 37.0 d |
| | | N180 | 2.31 b | 165.7 c | 241.4 c | 68.7 b | 169.8 c | 71.5 c |
| | | N240 | 2.34 b | 184.1 b | 270.3 b | 68.1 b | 185.1 b | 85.1 b |
| | | N300 | 2.48 a | 197.3 a | 299.5 a | 65.9 c | 201.6 a | 97.9 a |
| | | Mean | 2.31 A | 161.8 A | 236.8 A | 69.0 A | 163.9 A | 72.9 A |
| ANOVA | | | | | | | | |
| Year (Y) | | | ** | ** | ** | ** | ** | ** |
| Sowing (S) | | | ns | ** | ** | ** | ** | ** |
| N rate (N) | | | ** | ** | ** | ** | ** | ** |
| Y × S | | | ns | ns | ns | ns | ns | ns |
| Y × N | | | ns | ns | ns | ns | ns | ns |
| S × N | | | ** | ** | ** | ** | ** | ** |
| Y × S × N | | | ** | ** | ** | ** | ** | ** |

Notes:
** Significant at 0.01 probability level.
DS, drill sowing; WS, wide space sowing. Within a column for each growing season, means followed by different uppercase letters are significantly different according to Tukey's HSD test ($\alpha = 0.05$) between two sowing methods. Within a column for sowing method, means followed by different lowercase letters are significantly different according to Tukey's HSD test ($\alpha = 0.05$) among four N treatments. ns, not significant at 0.05 probability level.

The recorded NUE decreased with increases in N fertiliser administration. NUpE under WS was significantly higher by 11.98% and 10.15%, averaged across N rates, than that under DS in 2017–2018 and 2018–2019, respectively, whereas there was no significant difference in NUtE during the two growing seasons. The higher NUE under WS largely resulted from the advantage of NUpE over DS.

The AE$_N$ under WS was significantly higher by 16.51% and 21.05%, averaged across N application rates, than that under DS in 2017–2018 and 2018–2019, respectively (Table 7).

Table 7 N use efficiency (NUE), N uptake efficiency (NUpE), N utilization efficiency (NUtE), agronomic N use efficiency ($AE_N$), N recovery efficiency ($RE_N$), partial factor productivity of applied N ($PFP_N$) of winter wheat under different sowing method and N treatment in 2017–2018 and 2018–2019.

| Year | Sowing method | N treatment | NUE (kg kg$^{-1}$) | NUpE (%) | NUtE (kg kg$^{-1}$) | $AE_N$ (kg kg$^{-1}$) | $RE_N$ (%) | PFPN (kg kg$^{-1}$) |
|---|---|---|---|---|---|---|---|---|
| 2017–2018 | DS | N0 | 21.0 a | 63.9 a | 32.8 a | – | – | – |
| | | N180 | 16.7 b | 59.3 b | 28.1 b | 13.5 a | 55.4 a | 35.0 a |
| | | N240 | 15.0 c | 55.4 c | 27.1 b | 12.1 b | 49.8 b | 28.3 b |
| | | N300 | 11.5 d | 51.8 d | 22.3 c | 7.0 c | 45.5 c | 20.0 c |
| | | Mean | 16.0 B | 57.6 B | 27.6 A | 10.9 B | 50.2 B | 27.8 B |
| | WS | N0 | 24.4 a | 71.7 a | 34.0 a | – | – | – |
| | | N180 | 18.1 b | 65.1 b | 27.7 b | 13.5 a | 59.5 a | 38.1 a |
| | | N240 | 16.5 c | 61.9 c | 26.7 b | 13.4 a | 55.6 b | 31.8 b |
| | | N300 | 14.5 d | 59.3 d | 24.5 c | 11.2 b | 52.9 b | 26.0 c |
| | | Mean | 18.4 A | 64.5 A | 28.2 A | 12.7 A | 56.0 A | 32.0 A |
| 2018–2019 | DS | N0 | 26.4a | 76.1 a | 34.7 a | – | – | – |
| | | N180 | 19.0 b | 65.1 b | 29.2 b | 13.6 a | 55.2 a | 41.5 a |
| | | N240 | 17.8 c | 61.5 c | 29.0 b | 12.6 b | 51.6 b | 33.5 b |
| | | N300 | 14.0 d | 57.2 d | 24.6 c | 8.1 c | 46.9 c | 24.9 c |
| | | Mean | 19.3 B | 65.0 B | 29.4 A | 11.4 B | 51.3 B | 33.3 B |
| | WS | N0 | 29.1 a | 83.8 a | 34.7 a | – | – | – |
| | | N180 | 21.0 b | 70.5 b | 29.7 b | 15.1 a | 58.5 a | 45.7 a |
| | | N240 | 19.5 c | 67.2 c | 29.1 b | 14.4 b | 55.9 b | 37.3 b |
| | | N300 | 17.2 d | 64.8 d | 26.6 c | 12.1 c | 54.5 b | 30. 4 c |
| | | Mean | 21.7 A | 71.6 A | 30.0 A | 13.8 A | 56.3 A | 37.8 A |
| ANOVA | | | | | | | | |
| Year (Y) | | | ** | ** | ** | ** | ** | ** |
| Sowing (S) | | | ** | ** | ns | ** | ** | ** |
| N rate (N) | | | ** | ** | ** | ** | ** | ** |
| Y × S | | | ns | ns | ns | ns | ns | ns |
| Y × N | | | ns | ns | ns | ns | ns | ns |
| S × N | | | ** | ** | ** | ** | ** | ** |
| Y × S × N | | | ** | ** | ** | ** | ** | ** |

Notes:
** Significant at 0.01 probability level.
DS, drill sowing; WS, wide space sowing. Within a column for each growing season, means followed by different uppercase letters are significantly different according to Tukey's HSD test (α = 0.05) between two sowing methods. Within a column for sowing method, means followed by different lowercase letters are significantly different according to Tukey's HSD test (α = 0.05) among four N treatments. ns, not significant at 0.05 probability level.

$AE_N$ significantly decreased when the N application rate increased from 180 to 240 kg ha$^{-1}$ and then to 300 kg ha$^{-1}$ under both sowing methods and during the two growing seasons, except that there was no significant difference between 180 and 240 kg ha$^{-1}$ under WS in 2017–2018.

A higher $RE_N$ was observed under WS than that under DS by 11.55% and 9.75%, averaged across N application rates, in 2017–2018 and 2018–2019, respectively (Table 7). When the N application rate increased from 180 to 240 kg ha$^{-1}$ and then to 300 kg ha$^{-1}$,

 

$RE_N$ under DS significantly and continuously decreased, but did not decrease under WS in 2017–2018 and 2018–2019, respectively.

The wheat crop exhibited a significantly higher $PFP_N$ under WS by 15.11% and 13.51%, averaged across N application rates, than that under DS in 2017–2018 and 2018–2019, respectively (Table 7). With an increase in the N application rate, $PFP_N$ significantly and continuously decreased under both sowing methods and in the two growing seasons.

## DISCUSSION

The wheat crop under WS produced a higher grain yield than that under DS in the present study. The yield advantage under WS over DS was attributed to the increased ear number. Similar effects of WS on grain yield have been reported in previous studies (*Zhao et al., 2013*; *Fan et al., 2019*). The individual wheat plants under WS were more distant from each other than those under DS (*Zhao et al., 2013*), which reduced intraspecific competition for resources and growing space and resulted in more stems being produced (*Liu et al., 2017*; *Liu et al., 2020*). Additionally, WS significantly improved the productive stem percentage. It could be reasonably assumed that the wheat crop under WS could uptake more N and water from the soil, and then manufacture more carbohydrates through the canopy maintaining the growth and differentiation of the huge population of stems. It has been reported that WS can optimize root distribution and enhance the root absorptive capacity of wheat compared to DS (*He, 2020*). Our results also showed that, at the same N application rate, N uptake was higher under WS than that under DS.

An N input of 240 kg N ha$^{-1}$ is a locally recommended application rate which can produce high yields with acceptable NUE (*Chen et al., 2016*; *Zhang et al., 2016*; *Duan et al., 2019*). It is worth noting that the wheat crop under WS that were treated with 180 kg ha$^{-1}$ N produced a commensurate yield compared with that under DS with 240 kg N ha$^{-1}$ applied. This result indicates that an effective sowing method could compensate for the yield loss due to a 25% reduction in N input.

This study also found that, compared to the locally recommended N application rate (240 kg ha$^{-1}$), even though there was no significant increase in grain yield due to the higher N application (300 kg ha$^{-1}$) under WS, an evident increase of approximately 6% in grain N concentration was recorded. However, a significant yield reduction and no significant increase in grain N concentration were observed under DS with 300 kg N ha$^{-1}$ because stem lodging occurred during grain filling. It was reported that the wheat crop under DS with a high N application rate had a lower stem bending resistance and a higher wheat lodging possibility because of the contradiction between population and individual plants (*Liu, Bian & Liu, 2021b*; *Li et al., 2022*), which caused a decline in grain yield and quality (*Foulkes et al., 2011*). These results indicated that, even though a large amount of N fertilizer was applied, the enhanced sowing method, WS, could not only maintain a high yield level but also improve grain nutrient quality.

Crop yield was determined by dry matter production and harvest index (*Yoshida, 1972*). Thus, crop yield can be enhanced by increasing the dry matter accumulation, harvest index or both (*Rivera-Amado et al., 2019*). In the present study, higher total dry weight at maturity was achieved under WS than that under DS, whereas no significant difference in
harvest index was recorded between the two sowing methods. This result suggests that the wheat crop under WS exhibited greater total dry weight instead of higher biomass partitioning efficiency compared to those under DS. Additionally, pre-anthesis dry matter production and post-anthesis dry matter production were both significantly higher under WS than those under DS. Similar results were reported in a previous study (*Liu et al., 2017*). Dry matter production largely depends on the canopy photosynthetic area and leaf area index (*Man, Yu & Shi, 2017*; *Fan et al., 2019*), meanwhile leaf area growth is affected by N supply and uptake (*Peng & Ismail, 2004*). In the present study, the wheat crop under WS exhibited greater N uptake before anthesis than that under DS, which was attributed to a higher leaf area index at anthesis under WS over DS. It has also been reported that WS could increase the photosynthetic rate, delay senescence of flag leaves, and promote carbohydrate accumulation after anthesis (*Fan et al., 2019*). Our results also showed that wheat crops under WS that received 180 kg N ha$^{-1}$ could produce statistically equal amounts of dry matter (including pre-anthesis dry matter production and post-anthesis dry matter production, and total dry weight at maturity) compared to those under DS that received 240 kg N ha$^{-1}$, accounting for the same yield under the above-mentioned treatments.

In the present study, the total N quantity at maturity and N harvest index under WS were significantly higher than those under DS, which resulted in a higher grain N content under WS. The main source determining total N at maturity was the amount of N absorbed before and after anthesis (*Dupont & Altenbach, 2003*). Our study showed that wheat crops under WS had both higher pre-anthesis N uptake and post-anthesis N uptake than those under DS. In addition, the values of pre-anthesis accumulated N and post-anthesis accumulated N, and total N quantity at maturity under WS with the N application rate of 180 kg ha$^{-1}$ were statistically equal to those under DS with 240 kg N ha$^{-1}$. These results suggested that the enhanced sowing method caused wheat crop to exhibit adequate N uptake without high N fertilizer input.

The ability to enhance N uptake is the first step in increasing the NUE (*Du et al., 2020*). Our results showed that NUE under WS was significantly higher than that under DS, and the higher NUE was the result of improvement in NUpE instead of NUtE. Previous studies have also suggested that the variation in NUE under WS was more closely associated with NUpE than with NUtE (*Chu et al., 2018*; *Liu et al., 2021a*). The higher AE$_N$ and RE$_N$ in crop plants indicated that crop plants exhibited efficient use of N fertilizer, which decreased the loss of N fertilizer (*Yang et al., 2017*). The present study also showed that AE$_N$ and RE$_N$ under WS were significantly higher than those under DS. These results indicate that the enhanced sowing method, WS, could not only produce high grain yield but also decrease the risk of N leaching, ammonia emission, and ground runoff, which could cause environmental pollution. It is widely recognized that N use-related efficiencies (*e.g.* NUE, NUpE, NUtE, RE$_N$, AE$_N$, and PFP$_N$) will decrease when the N application rate increased (*Yang et al., 2017*; *Duan et al., 2019*; *Manschadi & Soltani, 2021*). It is remarkable that all the above-mentioned efficiencies under WS with the N application rate of 240 kg ha$^{-1}$ were comparable to those under DS with 180 kg N ha$^{-1}$ applied. These results indicate

that enhancement in the sowing method not only maintained high NUEs but also output high grain yield under moderate N fertilizer input.

## CONCLUSIONS

Compared to DS, WS had a higher grain yield which was mainly attributed to the ear number. The increased grain yield under WS was dependent on improved total dry weight. The higher total N quantity under WS was attributed to NUpE, which led to a greater NUE. The wheat crop under WS received N application of 180 kg ha$^{-1}$ and produced commensurate yield, total dry weight, and total N quantity compared with those under DS with 240 kg N ha$^{-1}$ applied. Thus, WS with moderate N fertilizer input can help maintain high N-use efficiencies and output high grain yield.

### Funding

This work was financed by the Research Program Sponsored by State Key Laboratory of Sustainable Dryland Agriculture (in preparation), Shanxi Agricultural University (No. 202003-1), the Ministerial and Provincial Co-Innovation Centre for Endemic Crops Production with High-quality and Efficiency in Loess Plateau (No. SBGJXTZX-38), the Shanxi Research Fund for outstanding doctor (No. SXYBKY2020005), the Shanxi Agricultural University Scientific Research Fund (No. 2020BQ41), the Modern Agriculture Industry Technology System Construction (No. CARS-03-01-24), the "1331" Engineering Key Innovation Cultivation Team-Organic Dry Cultivation and Cultivation Physiology Innovation Team (No. SXYBKY201733), and the Shanxi University Technological Innovations Plan (No. 2021L171). The funders had no role in study design, data collection and analysis, decision to publish, or preparation of the manuscript.

### Grant Disclosures

The following grant information was disclosed by the authors:
State Key Laboratory of Sustainable Dryland Agriculture (in preparation), Shanxi Agricultural University: 202003-1.
Ministerial and Provincial Co-Innovation Centre for Endemic Crops Production with High-quality and Efficiency in Loess Plateau: No. SBGJXTZX-38.
Shanxi Research Fund for outstanding doctor: SXYBKY2020005.
Shanxi Agricultural University Scientific Research Fund: 2020BQ41.
Modern Agriculture Industry Technology System Construction: CARS-03-01-24.
"1331" Engineering Key Innovation Cultivation Team-Organic Dry Cultivation and Cultivation Physiology Innovation Team: No. SXYBKY201733.
Shanxi University Technological Innovations Plan: No. 2021L171.

### Competing Interests

The authors declare that they have no competing interests.

## Author Contributions

- Qiang Wang performed the experiments, analyzed the data, prepared figures and/or tables, authored or reviewed drafts of the article, and approved the final draft.
- Hafeez Noor conceived and designed the experiments, analyzed the data, authored or reviewed drafts of the article, and approved the final draft.
- Min Sun analyzed the data, prepared figures and/or tables, authored or reviewed drafts of the article, and approved the final draft.
- Aixia Ren conceived and designed the experiments, analyzed the data, authored or reviewed drafts of the article, and approved the final draft.
- Yu Feng performed the experiments, prepared figures and/or tables, authored or reviewed drafts of the article, and approved the final draft.
- Peng Qiao performed the experiments, prepared figures and/or tables, authored or reviewed drafts of the article, and approved the final draft.
- Jingjing Zhang conceived and designed the experiments, performed the experiments, analyzed the data, authored or reviewed drafts of the article, and approved the final draft.
- Zhiqiang Gao conceived and designed the experiments, authored or reviewed drafts of the article, and approved the final draft.

## Data Availability

The raw measurements are available in the Supplemental File.

## Supplemental Information

Supplemental information for this article can be found online at http://dx.doi.org/10.7717/peerj.13727#supplemental-information.

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
