# Peer review of "Wide space sowing achieved high productivity and effective nitrogen use of irrigated wheat in South Shanxi, China"

_PeerJ, doi:10.7717/peerj.13727_

## Round 0.1 · original submission · Minor Revisions

All the reviewers suggested minor revision. Please consider detailed comments from Reviewer #2.

Reviewer 1 has requested that you cite specific references. You may add them if you believe they are especially relevant. However, I do not expect you to include these citations, and if you do not include them, this will not influence my decision.

Reviewer 1 ·

Basic reporting

Clear and unambiguous, professional English used throughout the manuscript. Lieterature is OK.Figures and results are also fine.

Experimental design

This portion is fine.

Validity of the findings

OK

Additional comments

I have gone through the exhaustive article on Wide space sowing achieved high productivity and effective nitrogen use of irrigated wheat in south Shanxi, China.
1. The “Title” of the article is appropriate and there is no need of any change.
2. All the selected relevant aspects have been well placed under the respective sub-title of the article.
3. Abstract is not well written; it is only a mere conscript of the study. Better would be to give some introduction followed by the gap in knowledge, hypothesis, general results and then conclusion. The abstract is the only part of the paper that the vast majority of readers see. Therefore, it is critically important for authors to ensure that their enthusiasm or bias does not mislead the reader.
4. The introduction resembles that of a review article and not that of a research article. What’s the gap of knowledge? Which is the scope of the manuscript? What hypothesis have been made? The introduction should be revised accordingly.
5. Experimental section:. A more succinic yet complete writing should be done. Moreover the author state that a statistical analysis has been made. I believe that the authors should give more details about the analysis performed,
6.The scientific background of the topic is poor. In "Introduction" and "Discussion", the authors should cite recent references between 2016-2020 from JCR journals.
Fahad, S., Sönmez, O., Saud, S., Wang, D., Wu, C., Adnan, M., Arif, M., Amanullah. (Eds.), (2021e.) Engineering Tolerance in Crop Plants Against Abiotic Stress, First edition. ed, Footprints of climate variability on plant diversity. CRC Press, Boca Raton.
Fahad, S., Sonmez, O., Saud, S., Wang, D., Wu, C., Adnan, M., Turan, V. (Eds.), 2021b. Climate change and plants: biodiversity, growth and interactions, First edition. ed, Footprints of climate variability on plant diversity. CRC Press, Boca Raton.
Fahad, S., Sonmez, O., Saud, S., Wang, D., Wu, C., Adnan, M., Turan, V. (Eds.), 2021c. Developing climate resilient crops: improving global food security and safety, First edition. ed, Footprints of climate variability on plant diversity. CRC Press, Boca Raton.
Fahad, S., Sönmez, O., Saud, S., Wang, D., Wu, C., Adnan, M., Turan, V. (Eds.), 2021a. Plant growth regulators for climate-smart agriculture, First edition. ed, Footprints of climate variability on plant diversity. CRC Press, Boca Raton, FL.
Fahad, S., Sönmez, O., Turan, V., Adnan, M., Saud, S., Wu, C., Wang, D. (Eds.), 2021d. Sustainable soil and land management and climate change, First edition. ed, Footprints of climate variability on plant diversity. CRC Press, Boca Raton.
Fahad S, Saud S, Yajun C, Chao W, Depeng W (Eds.), (2021f) Abiotic stress in plants. IntechOpen United Kingdom 2021. http://dx.doi.org/10.5772/intechopen.91549
Fahad S, Hasanuzzaman M, Alam M, Ullah H, Saeed M, Ali Khan I, Adnan M. (Eds.) (2020) Environment, Climate, Plant and Vegetation Growth. Springer Nature Switzerland AG 2020. DOI: https://doi.org/10.1007/978-3-030-49732-3
Fahad S, Abdul B, Adnan M. (Eds.), (2018b) Global wheat production. IntechOpen United Kingdom 2018. http://dx.doi.org/10.5772/intechopen.72559

Reviewer 2 ·

Basic reporting

no comment

Experimental design

no comment

Validity of the findings

no comment

Additional comments

1. In the References section, the format of the References is wrong, please correct the references format.
Journal reference format: List of authors (with initials). Publication year. Full article title. Full title of the Journal, volume: page extents. DOI (if available).
Example journal reference:
Smith JL, Jones P, Wang X. 2004. Investigating ecological destruction in the Amazon. Journal of the Amazon Rainforest, 112:368-374. DOI: 10.1234/amazon.15886.
2. In the Discussion section, please mention how your results compare to “Effects of Reduced Application of Nitrogen Fertilizer on Yield Formation and Nitrogen Use Efficiency of Winter Wheat With Wide-Range Sowing” which was published very recently.
3. What do ** and ns mean need to be added in Table 4, 5, 6 and 7.

Annotated reviews are not available for download in order to protect the identity of reviewers who chose to remain anonymous.

·

Basic reporting

The manuscript by Wang and associates provides a thorough comparison of N-related parameters between two wheat sowing methods under various N applications.

The writing is globally clear, but a careful proofreading by a fluent English speaker is necessary, as many sentences are not correct.
Especially (but nor limited to), few sentences are not understandable: lines 273-274 (“Wheat crop … both growing season”) ; lines 296-298 (“With the increase in N rate … respectively”) ; lines 338-339 (“Crop yield was determined … ways to increase”).

The manuscript is well structured, with sufficient context and appropriate references. Clear figures, data and raw data are provided.
Mentions to supplementary file should be added in the Result part.

Experimental design

Research question is well defined and investigation is sound.
Methods are globally well described. Calculation methods and definitions of Harvest Index and Nitrogen Harvest Index should be added.

Validity of the findings

Results are carefully analyzed and conclusions are supported by the results.

Additional comments

Additional minor comments:
- Line 93: “of fertilizer N absorbed” should be replaced by “of supplied N fertilizer”
- Table 1 (soil analysis): how many replicates were done? Have all the sub-plots been tested?
- Figure 1 suggest that plant density was different between the sowing methods, whereas text state that it is similar. Please clarify this, maybe by correcting the figure.
- Lodging in DS with N300 should be clearly stated in Results part (probably at several places). Indeed, it affects most of the measured parameters, as well as the comparisons between the sowing methods (it affects the statistical analysis, as sowing methods are compared globally instead of per N-condition); this effect should clearly appear (especially lines 219-220, 248-249, 250-252 and 259-260).
- Lines 227-228, “The stimulating effects … those under DS”: I don’t understand what this sentence is describing, and don’t identify these effects on Table 4.
- Lines 232, 239: mistakes in Fig numbers.
- Table 6: I am surprised by the result of statistical analysis for GNC% between N180, N240 and N300 for WS in 2017-2018. Is the table correct?
- “N uptake at maturity” (and related phrasings) is misleading, as the uptake is defined as N quantity taken up during a unit of time. It should be replaced by “total N taken up during life cycle” or (better) “total N quantity at maturity”. “Pre-anthesis uptake” and “post-anthesis uptake” are fine, as they refer to a period of time, not to a timepoint. Note: N uptake at maturity stage is actually null, as the plant is dead and cannot take N up anymore.

---

## Round 0.2 · Minor Revisions

Thanks for the manuscript update and detailed answer. Two reviewers recommended accept this paper for publication. However third reviewer still has some comments demanding minor revision. Please check and update manuscript accordingly. Pay attention to the English spellchecking. Waiting resubmission of the revised manuscript.

Reviewer 1 ·

Basic reporting

ok

Experimental design

ok

Validity of the findings

ok

Reviewer 2 ·

Basic reporting

no comment

Experimental design

no comment

Validity of the findings

no comment

Additional comments

no comment

·

Basic reporting

No comment

Experimental design

No comment

Validity of the findings

No comment

Additional comments

I would like to thank the authors for their answers to my comments.
However, I am not fully satisfied by some of the answers:

Reviewer #3, comment #1: despite the improvement, the manuscript would still gain from a careful proofreading by a fluent English Speaker.

Reviewer #3, comment #5: please modify Figure 1, to make clear there is the same plant density between sowing methods. Indeed, some readers will not fully read the Materials and Methods section, and this figure would thus be misleading.

Reviewer #3, comment #6: many measured parameters were affected in the 2 points where lodging occurred (DS with 300 kgN/ha in 2017-2018 and 2018-2019). The authors pinpointed this without mentioning the lodging : grain yield (lines 213-214), total dry weight at maturity (lines 243-244), harvest index (lines 246-248), pre-anthesis and post-anthesis dry matters (lines 256-257), grain N content (lines 268-269). As the lodging might be the cause of these observations, it should be mentioned in most of these paragraphs.

---

## Round 0.3 · Minor Revisions

Thanks for the updates and answer to the reviewer.

We have no remarks from the reviewers, but Section Editor has some comments requiring minor revision:

"This manuscript and its results are nicely done. There is one thing that needs clarification in Figures 2, 3, 4. From the legend "Within each growing season, bars with different upper-case letters are significantly different according to Tukey’s HSD test (α=0.05)". I can't figure out which bars are being referred to. There is an overall line with "B" on the left and "A" on the right, and the line covers to the two right bars (240 and 300) in the DS treatment and the two left bars (0 and 180) in the WS treatment. I can't tell what is A and what is B from this. Can you please ask the authors to clarify?"

Please answer and update the text accordingly to make it clear for journal readers.

---

## Round 0.4 · accepted · Accept

Thanks for the answer and update. I have no more comments.